# In Vitro and In Silico Studies for the Identification of Potent Metabolites of Some High-Altitude Medicinal Plants from Nepal Inhibiting SARS-CoV-2 Spike Protein

**DOI:** 10.3390/molecules27248957

**Published:** 2022-12-15

**Authors:** Saroj Basnet, Rishab Marahatha, Asmita Shrestha, Salyan Bhattarai, Saurav Katuwal, Khaga Raj Sharma, Bishnu P. Marasini, Salik Ram Dahal, Ram Chandra Basnyat, Simon G. Patching, Niranjan Parajuli

**Affiliations:** 1Center for Drug Design and Molecular Simulation Division, Kathmandu 44600, Nepal; 2Central Department of Chemistry, Tribhuvan University, Kathmandu 44618, Nepal; 3Department of Chemistry, Oklahoma State University, Still Water, OK 74078, USA; 4Paraza Pharma, Inc., 2525 Avenue Marie-Curie, Montreal, QC H4S 2E1, Canada; 5Nepal Health Research Council, Ramshah Path, Kathmandu 44600, Nepal; 6Oakridge National Laboratory, Bethel Valley Rd, Oak Ridge, TN 37830, USA; 7Independent Researcher, Leeds LS2 9JT, UK

**Keywords:** *Tinospora cordifolia*, spike protein, molecular docking, molecular dynamics simulation

## Abstract

Despite ongoing vaccination programs against COVID-19 around the world, cases of infection are still rising with new variants. This infers that an effective antiviral drug against COVID-19 is crucial along with vaccinations to decrease cases. A potential target of such antivirals could be the membrane components of the causative pathogen, SARS-CoV-2, for instance spike (S) protein. In our research, we have deployed in vitro screening of crude extracts of seven ethnomedicinal plants against the spike receptor-binding domain (S1-RBD) of SARS-CoV-2 using an enzyme-linked immunosorbent assay (ELISA). Following encouraging in vitro results for *Tinospora cordifolia*, in silico studies were conducted for the 14 reported antiviral secondary metabolites isolated from *T. cordifolia*—a species widely cultivated and used as an antiviral drug in the Himalayan country of Nepal—using Genetic Optimization for Ligand Docking (GOLD), Molecular Operating Environment (MOE), and BIOVIA Discovery Studio. The molecular docking and binding energy study revealed that cordifolioside-A had a higher binding affinity and was the most effective in binding to the competitive site of the spike protein. Molecular dynamics (MD) simulation studies using GROMACS 5.4.1 further assayed the interaction between the potent compound and binding sites of the spike protein. It revealed that cordifolioside-A demonstrated better binding affinity and stability, and resulted in a conformational change in S1-RBD, hence hindering the activities of the protein. In addition, ADMET analysis of the secondary metabolites from *T. cordifolia* revealed promising pharmacokinetic properties. Our study thus recommends that certain secondary metabolites of *T. cordifolia* are possible medicinal candidates against SARS-CoV-2.

## 1. Introduction

Natural products are the major source of antiviral drugs [1], which have been traditionally used as a remedy since ancient times and are proven to be effective. Aspirin, digitoxin, doxorubicin, quinine, and penicillin are some of the well-known natural product-derived drugs that possess anti-inflammatory, anti-diabetic, and several ethnopharmacological implications [2]. Several previous studies have reported that crude extracts from different medicinal plants such as *Tinospora cordifolia*, *Nelumbo nucifera*, *Glycyrrhizae uralensis*, *Pyrrosia lingua*, *Mollugo cerviana*, *Houttuynia cordata*, *Polygonum multiflorum*, and *Lycoris radiata* have promising results against coronaviruses [1,3]. In addition, a recent in silico study on the secondary metabolites of the plant *Andrographis paniculata* suggested that one of its major phytoconstituents, andrograpanin, could inhibit the main protease (M^pro^) of SARS-CoV-2 [4].

Enveloped coronaviruses (CoVs) with a genome size of 30–32 kb are large, spherical, or pleomorphic viruses broadly distributed among various mammals including humans, canines, felines, and also birds, and are the causative agents of a wide range of respiratory, hepatic, enteric, and neurologic infections [5,6,7]. Out of four genera of CoVs—alpha-, beta-, gamma-, and delta-CoV (α-, β-, γ- and δ-CoV)—α-Cov and β-CoV are associated with causing contagious and sometimes fatal respiratory infections in mammals. Similarly, γ and δ-CoV are reported in certain occurrences of avian infection [8]. Numerous CoVs (CoV OC43, canine CoV, 299E, porcine CoV, bovine CoV, etc.) related to humans and animals have been reported in the past thirty years; however, it was not until the SARS outbreak in 2002 and 2003 that CoVs were recognized as a potential human pathogen [9,10,11]. The year 2019 saw another CoV outbreak, this time on a much larger scale, and by the new pathogen SARS-CoV-2. COVID-19 caused by SARS-CoV-2 is a severe health issue because of its spread around the globe and the superior adaptation of this pathogen in human host cells in contrast to other CoVs that cause respiratory infections, such as SARS-CoV and the Middle East Respiratory Syndrome CoV (MERS-CoV). There are also a lack of proper antiviral drugs to combat COVID-19 [12]. Apart from those mentioned above, seven other types of CoVs are responsible for viral infections in humans, as in cases of the common cold through infection by HCoV-OC43, HCoV-NL63, HCoV-229E, or HCoV-HKU [13].

The genome sequence of SARS-CoV-2 shows ~80% sequence similarity with SARS-CoV and ~50% with MERS-CoV [14]. The genomic structure of SARS-CoV-2 is shared with the β-CoV group [15] and has 96.2% genomic similarity with bat CoV RaTG13. Therefore, bats are believed to be a primary host of SARS-CoV-2, which is then transmitted to humans via multiple transitional hosts [16]. SARS-CoV-2 primarily enters the human body via the respiratory tract and is transmitted through an infected patient’s fomites and from droplets while coughing or sneezing [17]. SARS-CoV-2 consists of four structural proteins, namely: spike (S), envelope (E), membrane (M), and nucleocapsid (N). In general terms, the S-protein facilitates virus entry to human cells through the ACE2 receptor [18,19,20], the M-protein is involved in virus assembly [21,22,23], and the E-protein contributes to the budding and release of progeny virus [24,25]. These membrane proteins contain all the features of entry, assembly, and release mechanisms inside the human cells [22,26]. Thus, the M-protein, S-protein Receptor Binding Domain (RBD) region, and E-protein transmembrane domain (TMD) region are some potential anti-SARS-CoV-2 drug target domains for effective antiviral therapy.

Scientists have been studying various existing medications that could be repurposed to cure COVID-19 patients. Both antiviral, as well as non-antiviral drugs, are repurposed to combat COVID-19. Several vaccines, such as mRNA-1273, BNT 162B2, Sputnik-V, NVX-CoV2373, and ZF2001 are licensed for use against COVID-19 [27,28]. On the other hand, SARS-CoV-2 continues to evolve globally, generating novel variants (variants of interest: B.1.525, B.1.526, B.1.526.1, B.1.617, B.1.617.1, B.1.617.2, B.1.617.3, P.2; variants of concern: B.1.1.7, B.1.351, B.1.427, B.1.429, P.1, B.1.1.529; and the variant of high consequence) [29] with changed transmissibility, infectivity, and coverage by therapeutics and vaccines [30]. Mutations in the S-protein of SARS-CoV-2 can change transmissibility, tissue tropism, and disease severity, such as the first spike mutation D614G that enhances spike binding to human ACE2 (hACE2) and enhances virus transmission [14]. It is therefore essential to closely surveil the consequences of emerging SARS-CoV-2 new variants in terms of virus transmissibility, infectivity, and efficacies of vaccines and emerging treatment regimens.

Moreover, various traditional medicinal plants are used to treat a wide range of viral and respiratory illnesses [31,32,33,34,35,36,37]. As an example, many people in Nepal and other developing countries have been using ethnomedicinal plants as medicine against various diseases for ages, and they have been found to be fruitful too. Since the incidence of the COVID-19 pandemic, people in these countries have been consuming such plants even more. However, there is no information about the nature and composition of the plants, or about their mechanism of action. Plant-active ingredients receive special attention since they have numerous structural and chemical properties that make them bioactive in combating various human ailments [1,38,39,40]. Compounds with antiviral activity have been found to alter or target several phases of the virus replication cycle, including adsorption, assembly, transcription, penetration, replication, uncoating, and release [41]. Hence, plant-based secondary metabolites could be an alternative source for screening antiviral drugs for their ability to block several pathways in the life cycle of SARS-CoV-2.

With the surge of COVID-19 all around the globe in quick succession, an efficient drug must be discovered quickly for its treatment and prevention. Scientists have repurposed drugs/vaccines which have previously shown antiviral effects against CoVs. Structure-based drug design can also be useful in combating COVID-19. A computational approach speeds up the drug discovery processes and plays a vital role in drug discovery, so the objective of this study was to seek potent compounds that inhibit the SARS-CoV-2 spike protein. The computational screening of compounds isolated from medicinal plants against a membrane protein, such as glycoproteins, receptor protein, or carrier protein of SARS-CoV-2, could generate baseline data for future in vitro and in vivo assays. Lan et al. [42] and Mandala et al. [43] observed the crystal structure of SARS-CoV-2 S1-RBD and the E-protein, respectively that proposed a new approach to developing SARS-CoV-2 inhibitors. In this work, we have carried out in vitro screening of crude extracts of seven ethnomedicinal plants thriving at high altitudes in Nepal against the S1-RBD of SARS-CoV-2, followed by in silico molecular docking of 14 reported antiviral compounds from *T. cordifolia* against the S1-RBD, providing new insights for future drug design studies. Further, molecular dynamics (MD) simulations of lead metabolites complexed with S1-RBD were performed to validate the results.

## 2. Results

### 2.1. In Vitro Screening

The crude methanolic extracts of seven ethnomedicinal plants (Table 1) were screened in vitro against the SARS-CoV-2 S1-RBD using an enzyme-linked immunosorbent assay (ELISA). A fixed concentration of all the plant extracts (5 mg/mL) was incubated with hACE2 in S1-RBD coated 96-well plates and a signal was detected using horseradish peroxidase (HRP)-linked secondary antibody. The percentage inhibition of the plant extracts was determined and displayed in Figure 1a along with their reported ethnomedicinal uses in Table 1. Interestingly, *T. cordifolia* crude extract was found to be better at blocking the interaction between hACE2 and the S1-RBD protein compared to the others. A 50% reduction in the binding of hACE2 with S1-RBD in the presence of *T. cordifolia* extract was achieved at approximately 1.25 mg/mL (Figure 1b). The detail of the measurement of absorbance and % of hACE2 bound to S1-RBD is shown in Appendix A. As a positive control, molnupiravir, included in the assay, showed 25% inhibition at a concentration of 50 µM. Overall, this method indirectly provided a shred of strong evidence that certain secondary metabolites in *T. cordifolia*, may be attributed to preventing the binding of hACE2 and SARS-CoV-2 S1-RBD, either individually or through their synergistic effect.

Following encouraging in vitro results that suggested that metabolites from *T. cordifolia* crude extract could prevent the interaction between hACE2 and SARS-CoV-2 S1-RBD, in silico studies of 14 reported antiviral compounds from *T. cordifolia* were conducted against the S1-RBD of SARS-CoV-2.

### 2.2. Molecular Docking of S1-RBD with Ligands

To gain insights into molecular recognition, drug discovery and medicinal chemistry can benefit from the use of molecular docking, which involves inserting small 3D structure ligands into the binding pocket of receptor structures. From the present investigation, it was observed that cordifolioside A (1), with the lowest negative S-score of −7.9942 and the highest GOLD fitness score of 58.27, binds competitively into the binding pockets of SARS-CoV-2 S1-RBD (Table 2). Similarly, palmitoside G (14) with an S-score of −7.1871 and a GOLD fitness Score of 50.80 interacts with the binding site residues. Additionally, amritoside B (10), cordifolide A (4), and palmitoside F (13) have shown better binding interactions with GOLD fitness scores of 50.08, 47.85, and 46.82, respectively.

Cordifolioside A (1) interacts through Thr430, Phe515, and Leu517 residues of S-protein RBD within a range of 2.36–2.83 Å. The protein–ligand interactions reveal four hydrogen bonds at the Thr430, Phe515, and Leu517 positions (Figure 2a). Palmitoside G (14) was also predicted to have hydrogen bond interactions with Arg355, Ser514, Phe 515, and Leu517 of the S1-RBD protein region (Figure 2b). The standard drug used as a reference, molnupiravir with an S-score of −2.9291, was found to interact with Arg346, Glu340, Val341, Asn354, Ser399, and Lys356 within a range of 2.84–4.73 Å in its keto hydroxylamine form [50]. Compared to the standard drug molnupiravir, both cordifolioside A and palmitoside G possess lower binding energy, which shows strong interaction of S1-RBD with them.

Moreover, our study further investigated and clarified the interaction between ligands and S1-RBD protein targets. GOLD fitness score values measure the prediction of binding affinities, docking accuracy, and speed poses. The 3D and 2D interaction of cordifolioside A with SARS-CoV-2-S1-RBD are depicted in Figure 2c and 2d, respectively. The details of the GOLD fitness score, binding energies, and interacting residues with the bond length of ligands to the S1-RBD protein region are depicted in Appendix A. Appendix A shows the binding interactions and 2D and 3D interaction of palmitoside G with S1-RBD.

### 2.3. Molecular Dynamics Simulation Analysis

Among the analyzed compounds of *T. cordifolia*, cordifolioside-A and palmitoside-G showed good binding interactions with the S-protein RBD region. To further investigate the ligand–receptor interactions, MD simulations were performed on the top-scored docked complexes (cordifolioside A- S1-RBD) compared to free S1-RBD protein. To analyze MD trajectories, the root-mean-square deviation (RMSD) is an important calculation. The average distance caused by the dislocation of a chosen atom over the time of a protein–ligand complex can be measured using RMSD [51]. During the simulation, the C-alpha atoms in the amino acids were considered to analyze the stability. The root-mean-square fluctuation (RMSF) is a measure of the flexibility of a residue.

The backbone RMSD values of atoms referring to the S1-RBD complex were used to understand the stability of the MD trajectories. The comparison of RMSD for the S1-RBD -cordifolioside-A complex with free protein is shown in Figure 3a. For the S1-RBD -cordifolioside-A complex, the average RMSD fluctuations for the protein and ligand are <0.40 nm, reaching equilibrium after 10 ns. The RMSD of cordifolioside-A was found to be stable from 0 ns to 160 ns, but it deviated from 160 ns to 200 ns, probably due to the presence of hydrogen atoms. Despite this, the ligand and protein complexes were well within an RMSD value below 0.35 nm, suggesting that it is stable and the best compound in the dynamic environment. The RMSD of free protein reveals slight fluctuation from about 150 ns to 155 ns. The differences between maximum and minimum RMSD values can explain the fluctuation of the backbone of the S1-RBD region when bound with selected ligands.

The flexibility among the amino acid residues is obtained from the RMSF [52]. To explore the conformational flexibility of the leading active site during the simulation process, VMD software was used to calculate the RMSF of all the amino acids around the ligand at 1 nm. The last 10 ns trajectory of MD simulations was examined before calculating the RMSF values of the complexes. The residues around the ligand and their RMSF values compared with the starting structures are shown in Figure 3b and the values are listed (Appendix A). The RMSF for each residue surrounding the ligand is less than 0.1 nm in the cordifolioside A-S1-RBD complex, indicating stability during the MD simulation. This high degree of fluctuation is due to the greater flexibility of the individual chain present, but the ligand remains stable during the entire simulation, proving the stability of the binding site. In contrast, there was suppression in the loop region 360 to 380 in the case of cordifolioside A. However, in the loop region 460–480, the residues are highly flexible in cordifolioside A. The reason behind this may be due to the allosteric mechanism. Figure 3c shows the radius of gyration of free protein and cordifolioside complexes.

Hydrogen bonding interactions are important in the protein–ligand complexes. To explore hydrogen bonds, the intermolecular interactions of ligands in complex with the S1-RBD region during simulation were computed within the last 1 ns trajectory of the MD simulations. Table 3 illustrates the hydrogen-bonding analysis for S1-RBD complexes in MD simulations. The analysis revealed that there is the formation of one hydrogen bond in the S1-RBD-cordifolioside A complex. The residue Phe515 played a significant role in the hydrogen bonding with cordifolioside A. Overall, the MD simulations of the ligand complexes with the S1-RBD region display stability and flexibility under dynamic conditions, and the analysis supports the binding energy predictions.

### 2.4. Binding Free Energy (BFE) Analysis

The Prime Molecular Mechanics Generalized Born Surface Area (MM/GBSA) approach was applied to determine the strength of the complex by calculating the absolute BFE (ΔG_bind_) [53]. In consideration of BFE and affinity, S1-RBD analysis showed that the compounds cordifolioside A (1) and palmitoside G (14) were preferred with significantly lower binding energies (−25.09 and −21.23 kcal/mol, respectively) suggesting a stronger binding affinity of cordifolioside A (1) and palmitoside G (14) to the spike protein.

Thus, the molecular docking, conformational dynamics analysis, and BFE analysis suggest the steady binding of cordifolioside A (1) with S1-RBD of SARS-CoV-2.

### 2.5. Analysis of ADMET Profiles

The absorption, distribution, metabolism, excretion, and toxicity (ADMET) properties of antiviral compounds analyzed through the pkCSM web server are listed in Appendix A. Drug absorption depends on colon cancer cells (Caco2), human intestinal absorption (HIA), and skin permeability. In ADMET profiles, good absorption in the human intestine is signified when the intestinal absorption value is above 30%. A greater HIA indicates that the compounds are absorbed in the intestine with ease. The volume of distribution (VDss), CNS permeability, and blood–brain barrier (BBB) permeability characterize the in vivo distribution of various drugs in tissues. None of the metabolites could cross the BBB readily. Metabolism of the drug is best predicted on the CYP models for substrate and inhibitors. Amritoside C (11), and palmitoside-F (13) inhibited CYP3A4. All 14 metabolites showed low renal clearance values. Appendix A shows the ADMET properties of the major compounds of *T. cordifolia* (1-14) by the pkCSM server.

Toxicity prediction through Pro-Tox-II revealed that cordifolioside A (1) and cordifolioside B (2) were categorized under the harmful category with toxicity class 5. Similarly, amritoside C (11), cordifolide B (5), cordifolide C (6), and palmitoside F (13) were found to be harmful if swallowed and were categorized under toxicity class 4. Moreover, Appendix A shows the predicted toxicity of secondary metabolites inhibiting metabolic enzymes using ProTox-II.

## 3. Discussion

Polyphenols and tannins have hydroxyl groups responsible for forming hydrogen bonds and dative bonds that impact envelope proteins preventing the attachment of a virus to the host cell. Moreover, the essential oils of many plants contain lipophilic terpenoids, which may intercalate between the lipid bilayer of the envelope, non-specifically causing lysis of the membrane due to a change in fluidity. Likewise, alkaloids act as intercalating agents that stabilize double-stranded DNA, ultimately inhibiting or reducing its replication [54,55,56]. A previous study on SARS-CoV revealed that alkaloids interacted with spike and nucleocapsid proteins and blocked their expression. Anthraquinones, such as emodin, and polyphenols, such as quercetin, tannic acid, and 3-isotheaflavin-3-gallate, impede the function of 3-chymotrypsin-like protease (3CL^pro^) along with adsorption and penetration. Various metabolites present in *Tinospora sp.* have been found to enhance the phagocytic activity of macrophage cells and build up an immune system by increasing white blood cells (leucocytes), which could be key to fighting infections [57]. Therefore, it is believed that plant-based secondary metabolites have similar modes of action against SARS-CoV-2. The similarity related to the progression of symptoms and mode of infection is also considered.

*T. cordifolia* has shown diversified ethnobotanical actions in which the plant is used to improve the body’s resistance to infections as an immune modulator [58]. Herbal medicine plays a significant role in preventing and controlling infectious virus diseases to decrease morbidity and mortality by enhancing host immunity against COVID-19 attacks [35,59,60,61,62,63]. A list of major natural compounds of *T. cordifolia* is shown in Appendix A.

Molecular docking is a theoretical and robust tool to predict the interaction between the target (receptor) and ligand, the activities of ligands still need to be confirmed by activity assays [64]. We used the molecular docking method to predict binding sites and the potential activity of top-scored compounds, which may contribute to discovering new potential inhibitors of the SARS-CoV-2 S1-RBD. The results were evaluated according to the docking poses and the ligand–protein interactions [65]. The low MOE S-score and the high GOLD fitness score of ligands lead to a better binding capacity to the protein residue [66]. Similarly, hydrogen bonding and hydrophobic interactions determine the binding affinity of protein–ligand complexes [67], and the lower the binding energy, the higher the binding affinity and stability of the complexes [68].

For S1-RBD, the interaction of cordifolioside A (1) to the binding pockets, through hydrogen bonds with Thr430, Phe515, and Leu517, is probably due to competitive inhibition. Moreover, cordifolide A (4), amritoside B (10), and palmitoside G (14) showed suitable GOLD fitness scores. Generally, the greater the GOLD fitness score, the better the docking result, and the stronger the protein–ligand interaction [66].

Furthermore, the BFE estimated from the final 10 ns MD simulation using MM/GBSA revealed that the complex was very steady for long-term simulation in the spike protein. The MD simulations indicated conformational differences in the binding site loop areas, which, when accounted for, provide a powerful, promising compound against the SARS-CoV-2 S1-RBD. Small compounds that target these binding pockets should interact with residues; disrupting the development of the oxyanions hole can result in its suppression [69]. The best potential drugs against SARS-CoV-2 can be analyzed by interpreting various thermodynamic parameters, such as density, temperature, pressure, RMSD, RMSF, hydrogen bond interaction, and hydrophobic interactions with their analysis plots [70]. Cordifolioside A (1) and palmitoside G (14) had the best docking poses, indicating that they have the potential to be exploited as potent drug candidates. Further, molecular docking and MD analysis of S1-RBD revealed that cordifolioside A had lower binding energy and stronger non-bonded interaction capability than the commercial drug molnupiravir and other compounds.

Intestinal absorption values, VDss, BBB permeability, and cytochrome P450 are some key parameters that need to be considered before the uptake of any drugs [71]. Compounds with a logBBB value of <−1 are poorly distributed in the brain, while those with a logBBB of >0.3 can cross the blood–brain barrier [72,73,74]. Similarly, the clearance value describes the amount of elimination of the drug and its concentration in the body [75]. Various other parameters, such as Ames toxicity [76], hepatotoxicity [77], and oral toxicity [78], play a key role in selecting drugs. Our computational investigations recommend using cordifolioside A of the genus *T. cordifolia* as a potent COVID-19 antiviral drug. However, due to the unavailability of cordifolioside A as a pure compound, we performed an in vitro study on *T. cordifolia* [79,80] crude extract of varied concentrations against S1-RBD of SARS-CoV-2, which revealed their inhibitory activity.

## 4. Materials and Methods

### 4.1. In Vitro Spike S1-RBD and hACE2 Inhibitory Activity of SARS-CoV-2 by Enzyme-Linked Immunosorbent Assay (ELISA)

Using a cold percolation, the chemical contents of the selected plants were extracted in methanol. To screen the inhibition efficiency of the crude extracts against S1-RBD –hACE2 binding, 5 mg/mL of each was incubated with hACE2 in S1-RBD coated 96-well plates (Novatein Biosciences, MA 01801, USA) and a signal was detected using an HRP-linked secondary antibody: Goat Anti-Human IgG-Fc, HRP conjugated (Novatein Biosciences, Woburn, MA, USA). For each extract, the assay was performed in triplicate, and % inhibition was expressed as a mean ± standard error of the mean of triplicates. Further, an increasing concentration of the most potent crude extract was added to a 96-well plate coated with recombinant S1-RBD [81] (catalog no. PR-ncov-2-PL, Novatein Biosciences, Woburn, MA, USA) and incubated for 2–4 h at 37 °C. Plates were washed in 3 × PBS pH 7.2 with 0.05% Tween-20 and blocked with 1% BSA and 0.05% Tween-20 in PBS. After three-times washing, 100 µL hACE2 receptor protein: Recombinant Human ACE2 (Met1-Ser740)/hFc Protein (catalog no.: PR-nCoV-4, Novatein Biosciences, Woburn, MA, USA) (0.1–0.2 µg/mL) was added and incubated for 1 h in binding buffer (0.1% BSA in PBS, pH 7.2). Plates were washed and 100 µL goat anti-human IgG-Fc, horseradish peroxidase (HRP) conjugated (1:500) in binding buffer was added as an enzymatic marker. After three washes, 3,3′, 5,5′-tetramethylbenzidine (TMB) (PerkinElmer Health Sciences Inc., Hayward, CA, USA) was added for signal generation, the reaction was stopped with an acidic solution, then the plates were read at 450 nm.

### 4.2. Computational Workstation

The molecular docking studies were performed on a Microsoft Windows workstation (Intel Core i7 processor and system memory 6GB RAM). The binding energy calculation was performed on an Intel Core-i7 processor, system memory of 8 GB RAM, and GPU Nvidia GeForce RTX 2060 6GB. MOE version 3.12 [82] was used for protein preparation and binding site analysis. GOLD version 4.0.1, based on a genetic algorithm, was used to examine the interactions of ligands with the target proteins [83]. Protein–ligand interactions were visualized on BIOVIA Discovery Studio [84], and final processing and graphical analysis were carried out using MOE.

### 4.3. Protein Preparation

The X-ray crystal structure of SARS-CoV-2 S1-RBD (PDB ID: 6M0J) [42] with a resolution of 2.45 Å was retrieved from the Protein Data Bank [85] and used for computational docking. Optimization was performed by removing water molecules, adding hydrogen atoms, and assigning atomic charges to all protein atoms using the standard preparation wizard, MOE [82].

### 4.4. Preparation of Ligands

A total of 14 secondary metabolites from *T. cordifolia* with reported antiviral activities were selected as ligands for molecular docking, which were accessed from PubChem and the literature [86]. The MOE ligand preparation module was used in preparing these ligands. Finally, the ligands were processed into mol2 file format and minimized for molecular docking using the MOE Lig-Prep module [87]. The chemical structures of the 14 secondary metabolites are depicted in Figure 4.

### 4.5. Binding Site Prediction

The Site-Finder module of MOE was used in determining the amino acids of the receptor involved in the formation of binding pockets [88]. Further, the S1-RBD binding site, size, and residues (Table 4) were calculated using Site-Finder.

### 4.6. Molecular Docking and Validation

After the complete refinement of protein structures, the ligands were docked into the binding sites of the target protein using MOE and the energy was minimized using the MOE Lig-X module [89]. In MOE, protein–ligand binding affinities with all feasible binding configurations were evaluated according to a numerical value known as the S-score. The lowest S-score inhibitors are more likely to generate a significant protein–ligand interaction at certain active sites. The poses of molecules were obtained and scored using the ASE scoring function [90]. For validating docking results, the top-scored compounds were extracted from their original binding site and were re-docked into the same position using the default GOLD docking protocol. The lowest energy pose acquired during re-docking and the previous docking positions of the compound were superimposed, and its RMSD was calculated. The RMSD values calculated between the co-crystallized ligands and the docked poses was 1.2 Å for the S1-RBD protein. The prediction of binding modes was considered successful where the RMSD value was below 2.0 Å [91,92], which indicated the validity and efficiency of the docking method.

### 4.7. Molecular Dynamics Simulation

The ligand-binding site, which changes the conformation of the protein and its impact on the protein–ligand complex, was studied by using MD simulation [93,94]. To determine the stability and flexibility of protein–ligand interactions, the S1-RBD of SARS-CoV-2 and their respective top-scored natural compounds were subjected to MD simulation. GROMACS 5.4.1 with a CHARMM force field [95] was used for this purpose on a LINUX-based workstation. A simple point charge (SPC) water model was used to solvate the protein–ligand complex using a cubic box with periodic boundary conditions (PBC). By adding appropriate Na^+^ or Cl^−^ ions, the overall charge of the system was neutralized. For minimization and relaxation of the system, the NPT ensemble available within the GROMACS package was used. Throughout the MD simulations, the temperature and pressure were kept constant at 300 K and 1.01325 bar, respectively. The RMSD, RMSF, and hydrogen-bonding interactions were analyzed. The MD simulation was performed for 200 ns following the protocol where the integration time frame was set to 0.002 ps [96,97]. The MD trajectories were generated at every 10 ps interval. The MD analysis included the calculation of RMSD and RMSF, along with analyses of hydrogen-bonding interactions of the protein–ligand complexes. Furthermore, the results of RMSD values were retrieved and plotted using VMD molecular dynamics visualization tools [98].

### 4.8. Binding Free Energy Calculation

Using the Prime MM/GBSA module of the Schrodinger suite with the OPLS Force Field, the BFE of the protein–ligand complexes was analyzed [99] using the following equation [100]:∆G_bind_ = ∆E_MM_ + ∆G_Solv_ + ∆G_SA_(1)
where ∆G_bind_ is the BFE of the protein–ligand system, ∆E_MM_ is the minimized energy difference between the 6M0J–ligand complex and the sum of the free protein and inhibitor. ∆G_Solv_ is the GBSA solvation energy difference of the protein–ligand complex and the sum of the solvation energies for the free receptor and free inhibitors. ∆G_SA_ is the surface area energy difference between the complex and the sum of the surface area energy for the unliganded receptor and ligand. To prioritize the lead inhibitors, the Prime MM/GBSA method was used as a rescoring function. Both the BFE and the docking scores were considered to optimize for the selection of top metabolites.

### 4.9. Pharmacokinetics Study of Secondary Metabolites

The pharmacokinetics parameters (ADMET properties) of potential anti-COVID-19 compounds were predicted by using a cheminformatics tool, i.e., the pkCSM web server [101]. Pro Tox-II was used to assess the potential toxicity present in the secondary metabolites where the lethal dose (LD_50_) was classified as fatal (class 1 and 2), toxic (class 3), harmful (class 4 and 5), and non-toxic (class 6) [102]. In addition, a reliability value higher than 0.7 predicts the confidence score of the secondary metabolites [103].

## 5. Conclusions

An effective strategy integrating in vitro screening and in silico studies, molecular docking, MD simulation, BFE calculation, and ADMET analysis was developed to screen for SARS-CoV-2 S1-RBD inhibitors from secondary metabolites, which was demonstrated to be valid and practicable (Figure 5). Based on our strategy, certain secondary metabolites from *T. cordifolia* might act as potent S1-RBD inhibitors. This study provides the basis for further exploration of natural products in the intervention and prevention of COVID-19 for future clinical use. Moreover, with the properties of targeted computational strategy and accurate analysis, this protocol is supposed to be further served for an extended range to rapidly screen active constituents from mixtures, which will expedite the efficiency of drug discovery and development using a computational approach. With the limitation of proper medications available against SARS-CoV-2, our findings are expected to stimulate research interest to unfold new dimensions of drug discovery. Further immunochemistry studies will bring fresh insights into SARS-CoV-2 therapy. Additionally, in vitro ELISA assays of the individual pure metabolites are strongly advised to distinguish between the metabolite’s inhibitions from its synergistic effects with other secondary metabolites.

## Figures and Tables

**Figure 1 molecules-27-08957-f001:**
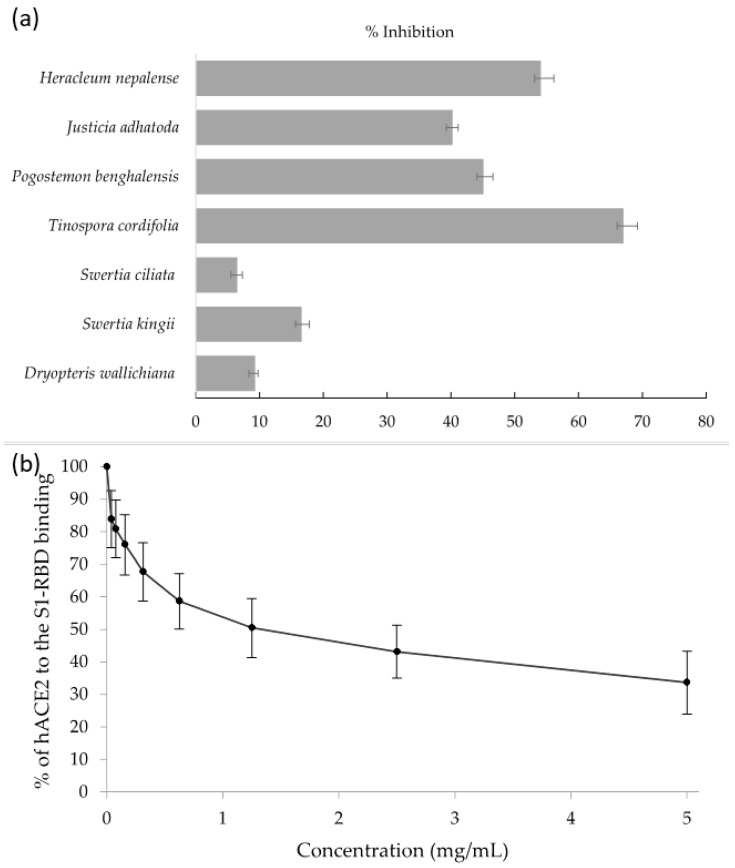
(**a**) Percentage inhibition of hACE2 binding with S1-RBD exhibited by plant extracts (5 mg/mL); (**b**) binding curve of hACE2 receptor to S1-RBD protein of SARS-CoV-2 in the presence of a range of crude extracts from *T. cordifolia* as determined by ELISA. The data represent mean ± SEM from *n* = 3 samples.

**Figure 2 molecules-27-08957-f002:**
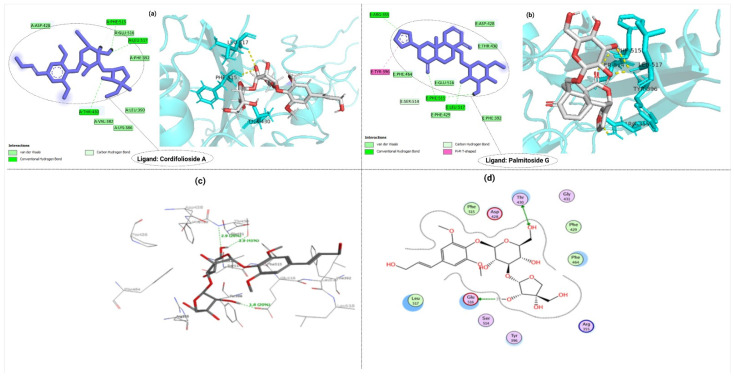
Interacting residues and the type of interactions of SARS-CoV-2 S1-RBD with (**a**) cordifolioside A and (**b**) palmitoside G; (**c**) 3D interaction of cordifolioside-A with SARS-CoV-2-S1-RBD; and (**d**) 2D interaction of cordifolioside A with SARS-CoV-2-S1-RBD obtained from Molecular Operating Environment (MOE).

**Figure 3 molecules-27-08957-f003:**
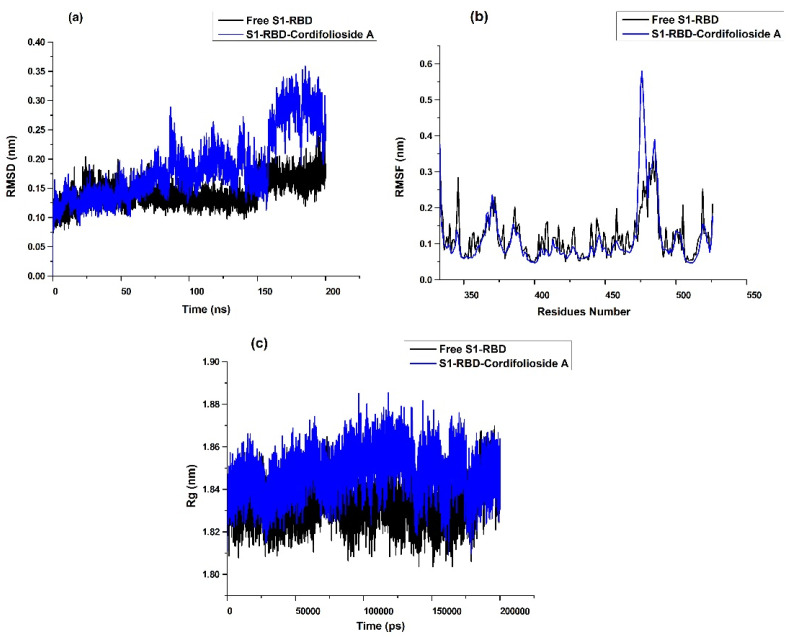
(**a**) RMSD of the atomic positions for the free S1-RBD and S1-RBD-cordifolioside A; (**b**) RMSF of the atomic positions for the free S1-RBD and S1-RBD-cordifolioside A; (**c**) radius of gyration for the free S1-RBD and S1-RBD-cordifolioside A for 200 ns MD simulation using GROMACS package.

**Figure 4 molecules-27-08957-f004:**
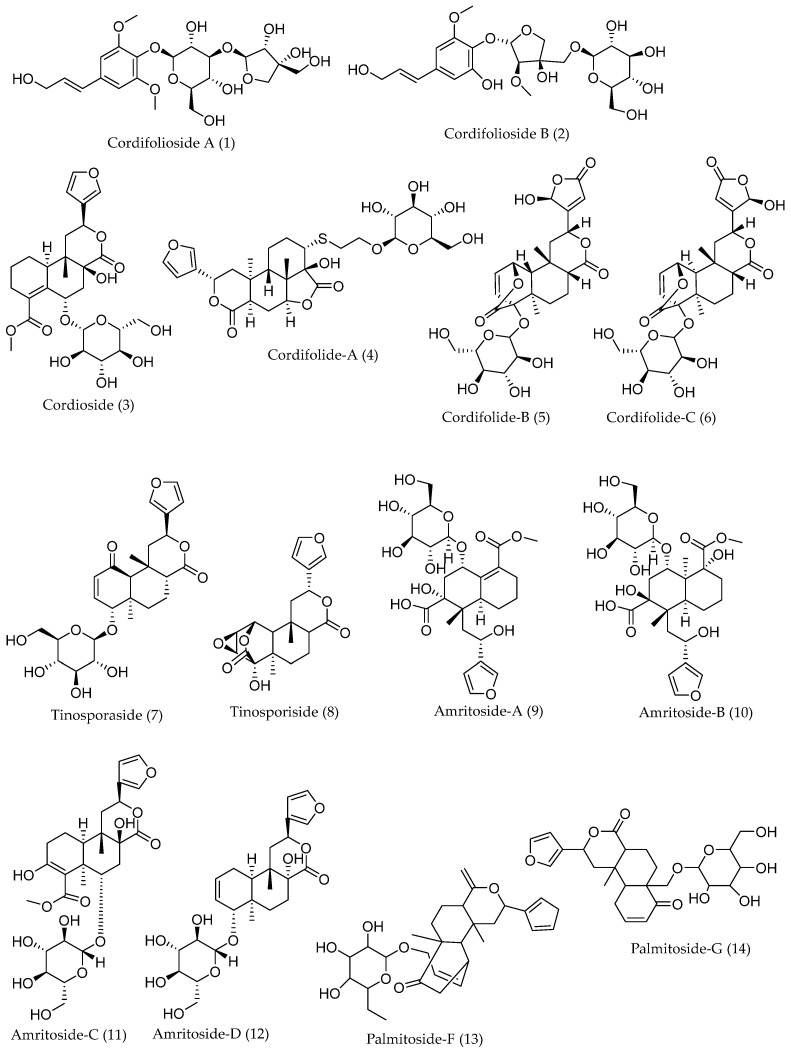
Chemical structures of secondary metabolites with antiviral activities from *T. cordifolia*.

**Figure 5 molecules-27-08957-f005:**
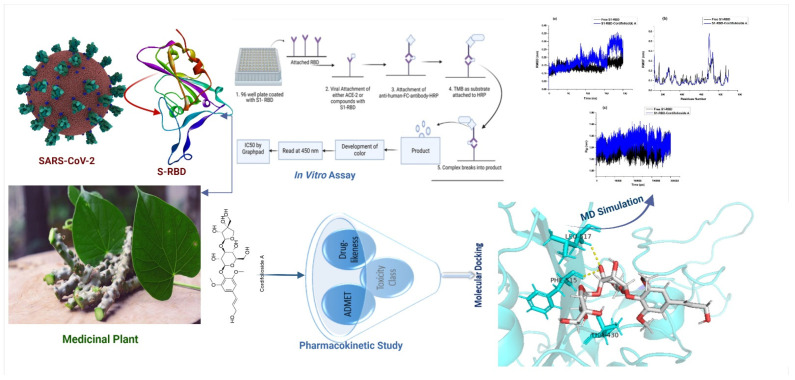
Workflow of the present study.

**Table 1 molecules-27-08957-t001:** Screening of crude extracts (5 mg/mL) of ethnomedicinal plants based on S1-RBD assay with their reported medicinal uses.

S.N.	Plants	Voucher Code	Location (Altitude)	Reported Medicinal Uses	References
1	*Dryopteris wallichiana*	KHP 03	Bajhang (2935 m a.s.l)	The rhizome is used as an anti-rheumatic and for treating constipation	[44]
2	*Swertia kingii*	KHP 08	Doti (3071 m a.s.l)	Blood purifier, skin disease, bitter tonic for fever, indigestion, laxative, anthelmintic, antidiarrhoeal, antiperiodic, and bronchial asthma	[45]
3	*Swertia ciliata*	KHP 24	Doti (3127 m a.s.l)	Used as a substitute for *Swertia kingii*	[45]
4	*Tinospora cordifolia*	TUCH 210052	Bajhang (2907 m a.s.l)	Immunomodulatory, anticancer, antiviral antidiabetic, antimicrobial, antioxidant, anti-inflammatory, antipyretic, and antiallergic	[46,47]
5	*Pogostemon benghalensis*	TUCH 210050	Bajura (3001 m a.s.l)	Antioxidant, anticancer, antibacterial, antifungal, anti-inflammatory, and antiviral	[47]
6	*Justicia adhatoda*	TUCH 210051	Doti (3107 m a.s.l)	Immunomodulatory, antimicrobial, antibacterial, antiviral, anti-inflammatory, and antioxidant	[47]
7	*Heracleum nepalense*	TUCH 210059	Bajura (3143 m a.s.l)	Breath rate stimulator, antidiarrheal, aphrodisiac, blood pressure stimulator, tonic, antioxidant, and antimicrobial	[48,49]

Note: (a.s.l.) = above sea level.

**Table 2 molecules-27-08957-t002:** MOE S-score, GOLD fitness score, binding free energy, and protein–ligand interactions of natural compounds with the SARS-CoV-2 S1-RBD region (6M0J).

Compound	S-Score	GOLD Fitness Score	Binding Free Energy MM/GBSA (ΔGbind) (kcal/mol)	Interacting Residues	Interaction Length (Å)
Molnupiravir	−2.9291	-	-	Arg346 Glu340 Val341 Asn354 Ser399 Lys356	2.84/2.99/4.36 2.05 4.95 4.16 3.07/2.40/1.96 4.73
Cordifolioside A (1)	−7.9942	58.27	−25.09	Thr430 Phe515 Leu517	2.36 2.40/2.83 2.65
Palmitoside G (14)	−7.1871	50.80	−21.23	Arg355 Tyr396 Ser514 Phe515 Leu517	2.37 4.92 3.26 2.83/2.90 1.91

**Table 3 molecules-27-08957-t003:** Hydrogen-bonding analysis for S1-RBD complexes in MD simulations.

S1-RBD-Complex	No. of Hydrogen Bonds	Interacting Residues	Bond Length (Å)	Hydrogen Bond Strength
S-Cordifolioside-A complex	1	Phe515	2.4	20%

**Table 4 molecules-27-08957-t004:** Binding site residues of the SARS-CoV-2 S1-RBD region (6M0J) identified using Site-Finder.

S1-RBD Binding Site	Size of Amino Acids	Residues
1	36	Arg454, Phe456, Arg457, Lys458, Ser459, Asp467, Ser469, Thr470, Glu471, Ile472, Tyr473, Gln474, Cyc480, Asn481, Gly482, Pro491
2	59	Arg355, Tyr380, Gly381, Val382, Leu390, Phe392, Tyr396, Pro426, Asp428, Phe429, Thr430, Gly431, Phe464, Leu513, Ser514, Phe515, Glu516
3	27	Arg403, Glu406, Lys417, Tyr453, Ser494, Tyr453, Ser494, Try495, Gly496, Phe497, Gln498, Asn501, Tyr505
4	18	Cyc336, Pro337, Phe338, Gly339, Phe342, Val367, Leu368, Ser371, Phe374

## Data Availability

Data are reported in the article and the Appendix A or are available from the corresponding authors upon reasonable request.

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
