# Peer review of "In Vitro and In Silico Studies for the Identification of Potent Metabolites of Some High-Altitude Medicinal Plants from Nepal Inhibiting SARS-CoV-2 Spike Protein"

_molecules, 2022, doi:10.3390/molecules27248957_

Round 1

Reviewer 1 Report

1. The title of the article should be revised like "In Vitro and In Silico Studies on Identification of Potent Metabolites of Some High-Altitude Medicinal Plants from Nepal Inhibiting SARS-CoV-2 Spike Protein".

2. The paper (DOI https://doi.org/10.1007/s11756-022-01012-y) related with medicinal plant and SARS-CoV-2 may be referred in introduction and discussion section, as similar work may strengthen your findings.

3. The work is good to report. A flow chart must be added in discussion section to clarify the steps.

4. Moderate English changes are required for clarity. 

Author Response

1. The title of the article should be revised like "In Vitro and In Silico Studies on Identification of Potent Metabolites of Some High-Altitude Medicinal Plants from Nepal Inhibiting SARS-CoV-2 Spike Protein".

Response: Thank you for the comment. The title of the article has been revised as suggested.

2. The paper (DOI https://doi.org/10.1007/s11756-022-01012-y) related with medicinal plant and SARS-CoV-2 may be referred in introduction and discussion section, as similar work may strengthen your findings.

Response: Thank you very much. The suggested paper has been referred to in the Introduction section as reference [4] on page 2.

3. The work is good to report. A flow chart must be added in discussion section to clarify the steps.

Response: Thank you very much. The workflow of the present study has been added in the Discussion section (Figure 4, page 10).

4. Moderate English changes are required for clarity. 

Response: Thank you for the comment. English language has been checked and edited.

Reviewer 2 Report

1) The activity of the representative compounds, such as cordifolioside-A and palmitoside-G is better to be tested in Vitro Enzyme-Linked Immunosorbent Assay (ELISA).

2) Docking Programs, such as Glide, in Schrodinger suite, Molecular Operating Environment (MOE) and BIOVIA Discovery Studio were not used, but Gold and autodock vina were used. Why? And Is there a uniform criterion for choosing a binding conformation from different docking programs?

3) Figure 3 and Figure 4 is better to be combined into a single graph.

4) In Molecular Dynamics Simulation Analysis, the RMSD of cordifolioside-A was found to be stable from 0 ns to 160 ns, but it deviated from 160 ns to 200 ns. So a longer time simulation is needed.

5) Since the Molecular dynamics simulation was done, MM/GBSA module was not used to calculate the binding free energy. But the Prime Molecular Mechanics Generalized Born Surface Area (MM/GBSA) module of the Schrodinger suite with the OPLS Force Field was used. Why?

6) Figure 6 was not clearly presented.

Author Response

1. The activity of the representative compounds, such as cordifolioside-A and palmitoside-G is better to be tested in Vitro Enzyme-Linked Immunosorbent Assay (ELISA).

Response: Thank you for the comment. We have performed in vitro testing on the crude extract of the plant Tinospora cordifolia. We are planning to carry out further research on the representative compounds soon.

2. Docking Programs, such as Glide, in Schrodinger suite, Molecular Operating Environment (MOE) and BIOVIA Discovery Studio were not used, but Gold and autodock vina were used. Why? And Is there a uniform criterion for choosing a binding conformation from different docking programs?

Response: Thank you for the comment. Although there are several docking programs, we have used MOE, GOLD, and Autodock Vina for docking. The complex obtained from MOE was then processed to GOLD software to obtain its GOLD score value and protein-ligand interactions were visualized using BIOVIA Discovery Studio. Similarly, Autodock vina was also used to calculate the desolvation energy of the complex.

Different docking programs are based on different algorithms. The docking algorithms provide several binding poses. The selection of the best docking poses is carried out based on the number of H-bonds present, hydrophobic interactions, interactions with the active sites, and binding energy values. These are some common criteria used in choosing binding conformations.

3. Figure 3 and Figure 4 is better to be combined into a single graph.

Response: Thank you for the suggestion. The original Figure 3 and Figure 4 have been combined into a single figure (Figure 2, page 8).

4. In Molecular Dynamics Simulation Analysis, the RMSD of cordifolioside-A was found to be stable from 0 ns to 160 ns, but it deviated from 160 ns to 200 ns. So a longer time simulation is needed.

Response: Thank you for the comment. Since both the free protein and complex showed an almost similar pattern of deviation, we did not carry out a simulation greater than 200 ns. Also, the ligand and protein complexes were well within an RMSD value below 1 Å, suggesting it is stable. This information was already given on page 7 of the manuscript.

5. Since the Molecular dynamics simulation was done, MM/GBSA module was not used to calculate the binding free energy. But the Prime Molecular Mechanics Generalized Born Surface Area (MM/GBSA) module of the Schrodinger suite with the OPLS Force Field was used. Why?

Response: Thank you for the comment. MM/GBSA module and Prime MM/GBSA module are the same. This has been corrected throughout the manuscript.

6. Figure 6 was not clearly presented.

Response: Thank you for the comment. To avoid ambiguity, Figure 6 has been removed from the manuscript.

Reviewer 3 Report

Review on molecules-1943010:

  Basnet, et al.: In Vitro and In Silico Study of Some High-Altitude Medicinal 2

  Plants from Nepal and Identification of Potent Metabolites In- 3

  hibiting SARS-CoV-2 Spike Protein

  It is probably useful to duscuss on the subject of the title in this way in journal Molecules. 

  It is not clear that data on Figure 1 was fully made by tha authors or taken from lterature, since it is a molecular dynamic study. 

  Which kind of enzymatic markers was used in ELISA method?

  In the Introduction I see many reason why the subject is important, but a liitle more soudb exlained why and how the system studied 

is chosen (The ligand-receptor interactions, etc.).

  All together, I can recommend this work to be published, the author should consider the comments above. 

Author Response

1. It is not clear that data on Figure 1 was fully made by the authors or taken from lterature, since it is a molecular dynamic study. 

Response:  Thank you for the comment. The data in Figure 1 was fully generated by the authors through in vitro screening experiments. In the revised manuscript, the original Figure 1 and Figure 2 are combined as Figure 1, where Figure 1(a) depicts the % inhibition of the crude methanolic extracts of seven ethnomedicinal plants and Figure 1(b) depicts the binding curve of the hACE2 receptor to the S1-RBD protein of SARS-CoV-2 in the presence of a range of crude extracts from Tinospora cordifolia as determined by ELISA. First of all, the selected plants were screened based on their S1-RBD/hACE2 binding inhibition resulting in T. cordifolia being the most potent plant. Then further in silico works were carried out on the secondary metabolites from T. cordifolia.

2. Which kind of enzymatic markers was used in ELISA method?

Response: Thank you for the comment. Horseradish peroxidase (HRP) was used as an enzymatic marker with 3,3’,5,5’-tetramethylbenzidine (TMB) as a substrate for signal generation in the ELISA method. The enzymatic marker has been mentioned in the manuscript on pages 3 and 12.

3. In the Introduction I see many reasons why the subject is important, but a little more should be explained why and how the system studied is chosen (The ligand-receptor interactions, etc.)

Response: Thank you for the comment. Many people in Nepal and other developing countries have been using ethnomedicinal plants as medicine against various diseases for ages and they have been found fruitful too. Since the incidence of the COVID-19 pandemic, people have been consuming such plants even more. As researchers, we were keen to know whether plants such as Tinospora species are scientifically useful or not. So, based on the findings of our study from both in vitro and in silico investigation, we found that Tinospora species is useful to lessen the effects of SARS-CoV-2. Further information has been added to the Introduction about the use of ethnomedicinal plants in Nepal (page 3). The manuscript already described how interaction of the SARS-CoV-2 spike (S) protein with the human ACE2 receptor is a key target for potential antiviral drugs.

Reviewer 4 Report

Authors have done a very basic experiments like molecular docking and dynamics. In vitro or in-vivo validation is required. After this, it can be resubmitted for consideration 

Author Response

1. Authors have done a very basic experiments like molecular docking and dynamics. In vitro or in-vivo validation is required. After this, it can be resubmitted for consideration.

Response: Thank you for the comment. In the present work, first of all, we performed in vitro testing on the crude extracts of seven ethnomedicinal plants against the spike receptor binding domain of SARS-CoV-2 using an enzyme-linked immunosorbent assay (ELISA). Following encouraging in vitro results for Tinospora cordifolia, in silico studies were conducted for the 14 reported antiviral secondary metabolites isolated from T. cordifolia. Further molecular docking and dynamics study revealed that Cordifolioside-A of T. cordifolia could act as an impending attenuator of the SARS-CoV-2 spike protein.

Round 2

Reviewer 2 Report

The author did not answer the reviewer's concerns well.

1. Cordifolioside-A and palmitoside-G is better to be tested in Vitro Enzyme-Linked Immunosorbent Assay, but not be planning to carry out further research on the representative compounds.

2. In Silico Studies, or Molecular Operating Environment (MOE) and BIOVIA Discovery Studio were used to prepare the proteins and small molecules for docking.  In docking study, Gold and autodock vina were used.  Why docking programs in Molecular Operating Environment (MOE) and BIOVIA Discovery Studio was not used?  

3. In Molecular Dynamics Simulation Analysis, the ligand and protein complexes were well within an RMSD value below 0.5 Å, suggesting it is stable. not 1 Å.

4. In fact, MM/GBSA module and Prime MM/GBSA module are not the same. 

Author Response

1) Cordifolioside-A and palmitoside-G is better to be tested in Vitro Enzyme-Linked Immunosorbent Assay (ELISA), but not be planning to carry out further research on the representative compounds.

Response: Thank you very much for the comment. We could only perform in vitro testing on the plant extract of Tinospora cordifolia following this paper DOI: 10.1021/acs.jpclett.0c03119. The promising in vitro data were further strengthened by the in silico results. The in vitro inhibition suggests that Tinospora cordifolia may inhibit the viral entry into the host cell and serve as a potential source of a lead compound. We are unable to purchase cordifolioside-A and palmitoside-G from Sigma-Aldrich at the current time. Thus, we are unable to perform in vitro ELISA with the pure compounds. It might take several months for us to isolate those molecules from Tinospora cordifolia. In our future research, we will be pursuing this matter.

2) In Silico Studies or Molecular Operating Environment (MOE) and BIOVIA Discovery Studio were used to prepare the proteins and small molecules for docking. In docking study, Gold and autodock vina were used. Why docking programs in Molecular Operating Environment (MOE) and BIOVIA Discovery
Studio was not used?

Response: Thank you very much for the comment. The main purpose of BIOVIA Discovery Studio is to visualize the protein and small molecules. Preparation of the protein and the ligand was performed using the MOE Lig Prep module. Thus, prepared protein and ligand were docked and several poses of complexes were generated using MOE. Based on the nature of H-bonding, type of H-bonding, and binding site residues, the best ligand poses were selected and then processed to GOLD software to obtain their GOLD fitness score value in their respective protein target. The higher the GOLD fitness score, the higher its stability. Similarly, to further evaluate the binding affinity of the selected ligands, Autodock vina was used. Summing up, BIOVIA is a visualization tool, and MOE, GOLD, and Autodock vina are docking tools used in this research study. The interaction images in Figure 2 are actually from docking performed by MOE and Autodock vina. This has been clarified in the manuscript.

3) In Molecular Dynamics Simulation Analysis, the ligand and protein complexes were well within an RMSD value below 0.5 Å, suggesting it is stable. Not 1 Å

Response: Thank you very much for the correction. This has been corrected in the manuscript (page 8, line 213).

4) In fact, MM/GBSA module and Prime MM/GBSA module are not the same.

Response: Thank you very much for the correction. There are two modules frequently used to calculate free energy - MM/GBSA and Prime MM/PBSA. Herein, we have used the Prime MM/GBSA module. This has been corrected in the manuscript (page 9, line 247; page 10, line 311; page 13, lines 432 and
431).

Reviewer 4 Report

Authors have replied to the comments and now it can be accepted.

Author Response

Thank you very much. 

Round 3

Reviewer 2 Report

Although the authors have provided a revised version, the manuscript has not been sufficiently improved to warrant publication. Crude extracts of Tinospora cordifolia bring a 50% reduction in the binding of hACE2 with S1-RBD in vitro. The result could not provided sufficient evidences for that cordifolioside-A, a major secondary metabolite in T. cordifolia, may be attributed to preventing the binding between hACE2 and SARS-CoV-2 S1-RBD, in particular in view of the synergies of different molecules. Cordifolioside-A (or palmitoside-G) is better to be tested in Vitro Enzyme-Linked Immunosorbent Assay (ELISA). It is very important for nest in silico studies. Another serious flaw is that MOE, GOLD, and Autodock vina were used in docking study,but the authors did not analyze and compare the results obtained by different docking methods, and based on what principle did the authors selected the best binding conformation among the many docking conformations. In table 2, GOLD fitness score and binding free energy from Autodock vina were presented. But in Figure 2,interaction model of SARS-CoV-2 S1-RBD with test compounds obtained from MOE were showed. The author seemed to have picked a result quite andomly.

Author Response

Dear Reviewer,

Thank you very much for providing this opportunity for revision. We have revised the manuscript as per the suggested comments. We have tried our best to address your comments as much as possible. We found the suggestions very positive and helpful for improving the manuscript. The following are the points in which the manuscript has been revised. We have highlighted the changed regions in green color in the manuscript.

Comment: Although the authors have provided a revised version, the manuscript has not been sufficiently improved to warrant publication. Crude extracts of Tinospora cordifolia bring a 50% reduction in the binding of hACE2 with S1-RBD in vitro. The result could not provided sufficient evidences for that cordifolioside-A, a major secondary metabolite in T. cordifolia, may be attributed to preventing the binding between hACE2 and SARS-CoV-2 S1-RBD, in particular in view of the synergies of different molecules. Cordifolioside-A (or palmitoside-G) is better to be tested in Vitro Enzyme-Linked Immunosorbent Assay (ELISA). It is very important for nest in silico studies. Another serious flaw is that MOE, GOLD, and Autodock vina were used in docking study,but the authors did not analyze and compare the results obtained by different docking methods, and based on what principle did the authors selected the best binding conformation among the many docking conformations. In table 2, GOLD fitness score and binding free energy from Autodock vina were presented. But in Figure 2,interaction model of SARS-CoV-2 S1-RBD with test compounds obtained from MOE were showed. The author seemed to have picked a result quite andomly.

Response: Thank you for your valuable comments. The manuscript has been amended to acknowledge the limitations of investigating the activity of crude extracts of Tinospora cordifolia, particularly in the Results and Conclusion sections. The observed activity of T. cordifolia has not been attributed to any particular secondary metabolites, rather it is inferred to certain metabolites in the crude extract of the plant or due to their synergistic effects. Further, it is also recommended to carry out an ELISA assay of individual pure compounds in the future.

Now, the manuscript has been revised to incorporate only the docking data from commercial software such as MOE and GOLD. Molecular docking data from the open-source AutoDock Vina have been removed to resolve ambiguity. Further, we have determined and updated the MOE S-score in the revised manuscript. It is a fact that the lower the MOE S-score and the higher the GOLD fitness score, the better the binding capacity/affinity of ligands to their respective protein targets. The ligands were chosen for further investigation mostly based on this (i.e. MD simulation). Therefore, there must be no indication that the outcomes were chosen at random.

Thank you very much

Corresponding authors